# Targeting Epigenetic Regulation of Cardiomyocytes through Development for Therapeutic Cardiac Regeneration after Heart Failure

**DOI:** 10.3390/ijms231911878

**Published:** 2022-10-06

**Authors:** Lindsay Kraus

**Affiliations:** College of Science, Technology, Engineering, Arts, and Mathematics, Alvernia University, Reading, PA 19607, USA; lindsay.kraus@alvernia.edu

**Keywords:** cardiac regeneration, neonatal cardiomyocytes, epigenetics

## Abstract

Cardiovascular diseases are the leading cause of death globally, with no cure currently. Therefore, there is a dire need to further understand the mechanisms that arise during heart failure. Notoriously, the adult mammalian heart has a very limited ability to regenerate its functional cardiac cells, cardiomyocytes, after injury. However, the neonatal mammalian heart has a window of regeneration that allows for the repair and renewal of cardiomyocytes after injury. This specific timeline has been of interest in the field of cardiovascular and regenerative biology as a potential target for adult cardiomyocyte repair. Recently, many of the neonatal cardiomyocyte regeneration mechanisms have been associated with epigenetic regulation within the heart. This review summarizes the current and most promising epigenetic mechanisms in neonatal cardiomyocyte regeneration, with a specific emphasis on the potential for targeting these mechanisms in adult cardiac models for repair after injury.

## 1. Introduction

Heart failure has been the leading cause of death worldwide for many years, making it a major public health and clinical concern globally [1]. Because of the detrimental and devastating effect heart failure induces on our population, it is crucial to understand the pathology and progression of the disease to find improved treatments and clinical approaches.

Current clinical approaches have been modest at best. Most treatments, including hypertensive medications, diuretics, and lifestyle changes, have helped decrease the risk factors of cardiovascular disease and heart failure [2], but there is currently no cure for heart failure [3]. To truly cure heart failure, the use of heart transplants and stem cells has been the major focus of most research [4,5]. More dire and aggressive measures are needed for heart failure treatment because the adult heart has a limited ability to regenerate after injury [6]. Specifically, the adult cardiomyocytes or muscle cells in the heart do not grow and divide frequently, leading to a loss of functional cells in the heart after an injury that is often replaced by scarring [7,8,9]. The use of stem cells was originally hypothesized as an option for cardiac cell replacement because of the ability of some stem cells to differentiate into cardiomyocyte-like cells and replaced any lost cardiomyocytes after injury. This is a major area of research currently, but engraftment issues, immune responses, and actual clinical approaches have still caused barriers to stem cell use in patients [10]. Thus, unfortunately, heart transplants are rare and stem cells have been less than promising. This has led to new cardiac-based therapeutics, including cardiac regeneration and epigenetic regulation of cardiac cells, specifically of cardiomyocytes [11]. 

Cardiac regeneration has been of interest because of the important timeline of cardiomyocyte development. The embryonic and neonatal mammalian heart has the ability to grow and repair after injury, but the adult mammalian heart has a subsequent inability to regenerate cardiomyocytes [12]. The embryonic mammalian heart can regenerate and grow cardiomyocytes in cycles of proliferation. Often, these new cardiomyocytes are derived from progenitor cells utilized during embryonic development [12,13]. The neonatal mammalian heart does have the ability to regenerate after injury. It is usually summarized that within the first week of life, neonatal murine cardiomyocytes can proliferate. More recent studies have narrowed this down to within the first two days of life, stating that neonatal mice can fully recover only after an aggressive injury that is received within the first two days after birth [14,15]. For other animals and models, this window of regeneration can vary [16,17]. However, for all mammals, once into adulthood, the ability of the heart to regenerate cardiomyocytes is lost, especially after severe injury [18]. Unlike mammals, the *Danio rerio* (zebrafish) is an exception to this ability. Zebrafish have cardiomyocytes that can regenerate into adulthood. Zebrafish can fully repair an adult heart after injury, making them a model organism to study cardiac regeneration [19]. 

Because there is such a distinct window of regeneration in most mammalian organisms, cardiomyocyte regeneration regulation has been a major area of focus in mouse, rat, and zebrafish models. Often, neonatal heart injury can be induced via apical resection, and a neonatal mouse heart can regenerate and heal to become fully functioning [20]. The ability is then lost into adulthood. Due to the regenerative potential of neonatal hearts, targeting neonatal regenerative genes and signaling has been considered for adult heart mechanisms, specifically after injury [21,22]. Current research has found that some developmental genes could be a major target [23,24,25], but the standout of these findings has narrowed major changes to epigenetic modifications [25,26]. 

Epigenetic modifications are chemical modifications that occur on top of the DNA, outside of the normal genetic coding [27]. Often, epigenetic regulation dictates chromatin structure and accessibility [28]. Epigenetic modifications and their subsequent chromatic regulation play a major role in compacting the DNA that is wrapped around histones within the nucleus of the cell, which can affect gene expression and DNA binding proteins [29]. These epigenetic modifiers often fall into three main roles: writers, readers, or erasers [30]. For example, an epigenetic modification such as methylation or acetylation can be added to a lysine residue, changing chromatin accessibility and, therefore, altering downstream gene expression. There are many types of epigenetic modification, which are summarized in this review. Specifically, these changes have been documented in many cardiac cell lines, including fibroblasts, endothelial cells, and cardiomyocytes [28,31]. Due to these changes, assessing the epigenetic mechanisms in neonatal versus adult cardiomyocytes would be of interest as potential targets in regulating cardiomyocyte regeneration as a possible therapeutic for heart failure.

## 2. Mammalian Cardiomyocyte Development and Differentiation

Cardiomyocytes are the main functional cells in the heart. They are responsible for the contraction and relaxation within the heart muscle [32], which has directly correlated these cells to heart function via echocardiography and histological measurements [33,34]. Often, with the loss of functional cardiomyocytes after injury, there is a decrease in heart function as measured by ejection fraction, fractional shortening, and cardiac output [34]. Because mammalian cardiomyocytes have a very specific ability to regenerate in the early stages of development, it is vital to understand the epigenetic differences and changes that occur in adult cardiomyocytes. Embryonically, cardiomyocytes are derived from the mesoderm during gastrulation. Specifically, these cardiac precursor cells form a cardiac crescent, which is often where committed cardiovascular cells are found during development [35]. The commitment to the cardiac lineage is often associated with transcription factors, such as Nkx2.5 and Gata4 [35]. Interestingly, more recently, this has also been connected to epigenetic changes, including increased histone modifications, DNA methylation, and chromatin remodeling, directly regulating cardiomyocyte differentiation genes and pathways [9]. In a neonatal heart, the cardiomyocytes have a unique ability to regenerate during the first few days after birth [20]. It has been found that earlier in the neonatal stages, the cardiomyocytes have increased ability to repair after injury, respond to immunological challenges, and even divide regularly often due to changes in chromatin accessibility and epigenetic regulation [25]. Targeting these mechanisms has been of interest for potential cardiomyocyte regeneration and will be discussed throughout this review. 

The neonatal mammalian cardiomyocytes eventually become terminally differentiated [36]. These are deemed the mature adult cardiomyocytes, which are known for the contractile function in the adult heart muscle. Once the cardiomyocytes become terminally differentiated, they can no longer regenerate. Importantly, this means they do not go through the cell cycle regularly [37]. Thus, neonatal cardiomyocytes are often categorized by genes that are associated with this dedifferentiated state or ability to transition through the cell cycle [38], while adult cardiomyocytes are often labeled and characterized by genes that are associated with this terminally differentiated state [39]. Additionally, adult cardiomyocytes are often characterized by their mitochondrial function, which can provide a detailed understanding of the adult cell metabolism [40]. The maturation of adult cardiomyocytes is associated with the cell’s terminal structure, metabolism, and function of differentiated cardiomyocytes [23]. Normally, the need for new mature cardiomyocytes is not necessary in the adult heart. However, after injury or stress, the heart loses adult cardiomyocytes and can no longer grow and replace the lost cells [41]. This results in the cells being replaced by proliferative pathological fibroblasts, which increase scarring and decreases heart function. Overall, this cellular replacement can have long-lasting negative effects on health and heart function [42]. Because of these monumental changes, targeting the differences between the differentiated adult cardiomyocytes and their neonatal counterparts have been of interest, with the overall goal of reprogramming adult cardiomyocyte to be like their neonatal counterparts [43]. One of the most substantial methods for this theory has been through epigenetic regulation in neonatal versus adult mammalian cardiomyocytes.

## 3. Epigenetic Regulation of Cardiomyocytes through Development

Assessing changes in epigenetic regulation through cardiomyocyte development has been of interest because of the drastic change in the regenerative ability of the heart after the first few days of life [15]. Changes in epigenetics have proven to vary from neonatal to adult differentiated cardiomyocytes and may be a novel therapeutic target for heart failure [9,25,30], and as summarized in Table 1.

### 3.1. DNA Methyltransferases (DNMTs)

The epigenetic regulation by DNA methyltransferases (DNMTs) has been studied in many models including zebrafish, mice, rats, and even humans [44]. This epigenetic modification often transfers a methyl group to a C5 position on cytosine. The methyl group addition can affect gene expression and binding of transcription factors directly to the DNA [45]. There are also scenarios in which the methyl group is removed, called demethylation which has the reverse effect [45]. Two well-studied DNMTs include DNMT3a and 3b, which have been shown to regulate various genes’ expression in cardiomyocytes through methyl-CpG binding domains, which alter chromatin accessibility and therefore downstream gene expression. Specifically, a study found that DNA methylation by these DNMTs can alter important transcription factors such as MeCP2, a methyl-CPG binding protein [46]. MeCP2 caused increased methylation in neonatal rat cardiomyocytes that are not seen in adult differentiated cardiomyocytes, thus making it a potential target for cardiomyocyte regeneration and differentiation mechanisms. Another study found that DNMT3a was essential for embryonic cardiomyocyte phenotypes, specifically that the loss of DNMT3a caused a decrease in morphology and contractility that was deemed vital during development [47,48]. Additionally, it was found that the inhibition of DNA methyltransferases, specifically using 5-azacytidine, caused a stunt in zebrafish cardiomyocyte development that parallels the changes seen in adult mammalian cardiomyocytes [49].

### 3.2. RNA Methylation

Similar to DNA, RNA molecules can be methylated, specifically messenger RNA (mRNA). A common mRNA epigenetic modification includes the methylation of adenosine at the N^6^ position (m^6^A) [50]. The role of m^6^A methylation in the development of cardiomyocytes is still not completely understood; however, this epigenetic modification has been initially associated with the progression of some cardiovascular disease states [51]. More specifically, this modification was enhanced in the development of cardiac hypertrophy of human and mouse cardiomyocytes [52,53]. It is believed that the m^6^A modification regulates the primary factor called methyltransferase-like 3 (METTL3), which is directly connected to cardiac disease progression [54]. Importantly, in neonatal cardiomyocytes, it was found that the m^6^A modification leads to the loss of miRNAs, such as miR-133a, leading to changes in cardiac development [55]. Another study found that the expression of the m^6^A epigenetic modification varies over the first week of life in rat models, with an increase in the modification at p0 that is lost by p7 [56]. Overall, this RNA epigenetic modification has a great influence on cardiomyocyte homeostasis and a major role in the development of functional cardiomyocytes [57].

### 3.3. Histone Methylation

Unlike DNA or RNA methylation, histone methylation involves changes to chromatin accessibility due to the addition or loss of a methyl group to a histone tail [58]. Epigenetic regulation by histone modifiers such as the Polycomb Repressive Complex 2 (PRC2) has been shown to modulate the epigenetics of cardiomyocytes through development [59]. A study found that Ezh2, the PRC2 histone modifier that methylates histone 3 at lysine 27, was used to transition progenitor cardiomyocytes to adult differentiated cardiomyocytes. Yue et al. found that a knockout of Ezh2 causes a loss of this epigenetic methylation and loss of neonatal cardiomyocyte proliferation. This study linked the epigenetic regulation in the heart to platelet-derived growth factor receptor β (PDGFRβ) and the phosphatidylinositol 3-kinase (PI3K) pathway [60]. Additionally, Tang et al. found Ezh2 as a block for reprogramming, and by inhibiting this epigenetic methylation it resulted in increased cardiac reprogramming of human inducible cardiomyocytes [61]. Finally, a study that assessed changes in H3K27me in zebrafish development found that the silencing of this epigenetic modification in adult models provided cardiac regeneration similar to those of neonatal mouse models [62]. The tri-methylation of histone 3 on lysine 9 (H3K9me3) has also been shown to have a profound effect on the development of mouse cardiomyocytes. This epigenetic modification was found to directly regulate the expression of a fetal troponin gene necessary for myofibril proteins in the heart [63]. Interestingly, this epigenetic mark has also been associated with mitochondria function and adult cardiomyocyte metabolism [64]. Other histone methylations, including the H3K4me3, have been associated with the terminally differentiated state of adult mammalian cardiomyocytes as well [65].

### 3.4. Histone Acetylation

Histone acetylation is a form of epigenetic regulation that has been strongly associated with increased gene transcription. Therefore, the loss or removal of the acetyl group has often been connected to a decrease in gene expression [66]. Regarding cardiomyocytes, histone acetylation has been linked to cardiomyocyte proliferation, differentiation, and regeneration [67,68]. Specifically, the acetylation of H3K9 was associated with a more hypertrophic response in heart failure and led to the progression of a cardiomyocyte disease state [67]. Another study found that histone acetylation linked with UCP2, a metabolic regulator, altered not only the chromatin but also the metabolism of cardiomyocyte cells. This was specifically linked to hypoxic conditions and the regenerative ability of neonatal cardiomyocytes in the early stages of development. In the hypoxic condition, there was an increase in histone acetylation, but without UCP2 that acetylation was lost, indicating a connection between cardiomyocyte growth and regeneration associated with cardiomyocyte metabolism [69]. Histone acetyltransferases (HATs) have been shown to regulate stem cell differentiation into cardiomyocytes. Specifically, HAT activation of Gcn5 was found to cause terminal differentiation of mesenchymal stem cells to differentiated cardiomyocytes [70].

### 3.5. Histone Deacetylation

Histone deacetylases (HDACs) have also been studied for their role in epigenetic changes in cardiomyocytes [68]. The general role of HDACs has been connected to the growth and regulation of neonatal mammalian cardiomyocytes [68]. HDACs are very diverse and have been classified into four main categories, class I, class IIa, class IIb, and class IV based on functions in various cell types [71]. Specifically, the class IIa HDACs, which include HDAC4, 5, 7, and 9 have been associated with the Mef2 gene, a vital regulator of the adult cardiomyocyte phenotype [71,72]. Additionally, the gene, Brg1, has been shown to maintain the fetal cardiomyocyte state, allowing for growth and cardiac gene expression. This Brg1 gene was specifically found to regulate HDAC mechanisms through development [73]. In zebrafish, it has been found that the histone deacetylates 1 (HDAC1) is conserved in zebrafish and plays a vital role in the regeneration and proliferation of cardiomyocytes similar to that of neonatal and embryonic mammalian cardiomyocytes [74]. Finally, it has been observed that by inhibiting HDACs, such as HDAC2, has helped with cardiac repair after injury by specifically inhibiting increased cardiac cell autophagy [75].

### 3.6. Histone Ubiquitination

Histone ubiquitination is a dynamic epigenetic modification, as it has many roles and functions modulating gene expression. Often, histone ubiquitination occurs on histones 2A, 2B, and 3. The most well-studied is the single ubiquitination addition on H2A at lysine 119 [76,77]. The addition or removal of a ubiquitin group has been associated with genomic stability, gene expression regulation, and DNA damage regulation in many cell types [78]. In cardiomyocytes, histone ubiquitination has been connected to heart development and even regulating the progression of congenital heart disease [79]. It has been found that mono-ubiquitination by the Polycomb Repressive Complex 1 (PRC1) has played a major role in maintaining cardiac profiles, transcriptional regulation, and DNA damage through development [80,81]. In a mouse model, the ubiquitination of histone 2a on lysine 120 was found to regulate the maturation of cardiomyocytes through the RNF20/40 complex [82], which had an overall effect on cardiomyocyte gene expression through development.

### 3.7. SUMOylation

One form of epigenetic regulation similar to histone ubiquitination is SUMOylation, or small ubiquitin-related modifier (SUMO), which is usually the addition of one ubiquitin modification to a histone [83,84]. SUMOylation has been connected to cardiomyocyte gene expression and development [85,86]. This epigenetic modification has been directly connected to cardiomyocyte-regulated proliferation through development using an E3 or E2 ubiquitin ligase [85,87]. The SUMO1 and SUMO2 were connected to the embryogenesis and the growth of cardiomyocytes [87]. SUMO1 was found to regulate the Nkx2-5 gene transcription in the cardiomyocyte development [88]. SUMO2 was found to target the PcG regulation of Gata4 in cardiomyocyte growth and development [87,89]. Specifically, this SUMOylation epigenetic mechanism was associated with cardiomyocyte renewal and disease progression regulation through the modulation by p65 in the inhibition of cardiomyocyte hypertrophy [90], which is summarized in Table 1.

**Table 1 ijms-23-11878-t001:** Summary of cardiomyocyte-related epigenetic modifications.

Epigenetic Modification	Species/Model	Summary of Role in CM Regeneration and Development	References
DNMTs DNMT1 DNMT2 DNMT3a/3b	Rat ventricular myocytes, mouse embryonic cardiomyocytes, zebrafish cardiomyocytes	Increased regenerative ability and proliferation, necessary for embryonic development, increased survival, increase CM gene expression	[46,47,48,49,91,92]
RNA Methylation m^6^A	Mouse and human adult cardiomyocytes, neonatal mouse and rat cardiomyocytes	Increased development of hypertrophic CMs, regulation of CM growth	[52,53,55,56]
Histone Methylation H3K27me3 H3K4me3 H3K9me3	Human inducible cardiomyocytes, zebrafish heart, mouse cardiomyocytes	Increased regenerative ability, improves cardiac reprogramming and cell growth, increases cardiac gene expression and metabolism	[60,61,62,63,64,92,93,94]
Histone Acetylation and Histone Acetyl Transferases (HAT) H3K9ac H3K14ac HAT	Adult and neonatal mouse cardiomyocytes	Regulation of cardiomyocyte regeneration, reprogramming, development	[69,93,94,95]
Histone Deacetylases (HDAC) HDAC1 Class IIa HDACs (4, 5, 7, 9)	Zebrafish heart, mouse cardiomyocytes	Needed for development and regeneration, Mef2 expression	[71,72,74]
Histone Ubiquitination H2Bub1 H2BK120ub H2AK119ub	Adult and neonatal mouse cardiomyocytes, *Xenopus* heart	Cardiac gene expression, gene transcription, and cardiomyocyte maturation	[79,80,82]
SUMOylation SUMO1/PIAS1 SUMO2	Neonatal Sprague–Dawley rats, neonatal and adult mouse cardiomyocytes	Cardiomyocyte survival, neonatal cardiac gene expression	[88,89,90]

## 4. Targeting Neonatal Epigenetic Mechanism in Adult Cardiomyocytes for Therapeutic Potentials following Heart Failure

Due to these changes seen in neonatal versus adult cardiomyocytes, targeting epigenetic regulation could provide a clinical approach to healing a failing heart. By targeting neonatal epigenetic regulation in adult cardiomyocytes, clinicians could regenerate previously thought to be terminally differentiated cardiomyocytes as illustrated in Figure 1. By reintroducing neonatal epigenetic mechanisms in adult cardiomyocytes, cardiac regeneration could be possible in pre-existing cells in the heart. It has been hypothesized that by targeting the neonatal pattern of epigenetic histone modifications and CpG sites of methylations, adult cardiomyocytes could become proliferative and regenerative [96]. Specifically, this has been attempted by targeting the enhancer region of cardiomyocytes after injury in an attempt to activate epigenetic regulation, such as the H3K27ac as well as DNA methylation at 5-cytosine [96].

Due to the potential of targeting more neonatal-like epigenetic regulation in adult cardiomyocytes, direct epigenetic-oriented drugs have been considered for a therapeutic approach to cardiovascular disease [97]. These “epidrugs” would potentially interfere with various epigenetic mechanisms to enhance regenerative cardiomyocytes or inhibit the negative response to cardiac injury [97]. It has even been suggested that some standard hypertensive drugs could play a major role in indirect epigenetic regulation [98]. Finally, the reconfiguration of mCpG regions has been suggested as a more individualized approach to cardiovascular-based therapies via epigenetic regulation [99].

## 5. Conclusions, Limitations, and Future Directions

With major phenotypic differences in neonatal versus adult cardiomyocytes, therein lies some powerful therapeutic potential to be utilized for heart failure treatments. Specifically, targeting epigenetic mechanisms commonly found in the neonatal cardiomyocytes and optimizing them in adult cardiomyocytes provides a promising and powerful tool for cardiac regeneration. Currently, there are modest changes that can be made to prevent adverse cardiac events, with no substantial cure for heart failure. Targeting neonatal epigenetic mechanisms in adult cardiomyocytes is therefore a novel approach to heart failure treatments. With that, there are limitations and barriers that still need to be understood and overcome to fully utilize epigenetic mechanisms in cardiomyocytes for heart failure therapy. There is still a need to understand the adverse effects of instigating neonatal epigenetic mechanisms and how that would affect other cells, organs, and immune responses. Epigenetic modifications have also been strongly associated with microRNAs, long non-coding RNAs, and circulating RNAs. There is extensive data that indicate a connection with these signaling molecules that would be of interest moving forward as well [100,101,102]. Overall, epigenetic regulation differences in neonatal and adult cardiomyocytes are currently the untapped potential of cardiac regeneration therapy, which could eventually provide a great resource and therapeutic for heart failure patients in the future.

## Figures and Tables

**Figure 1 ijms-23-11878-f001:**
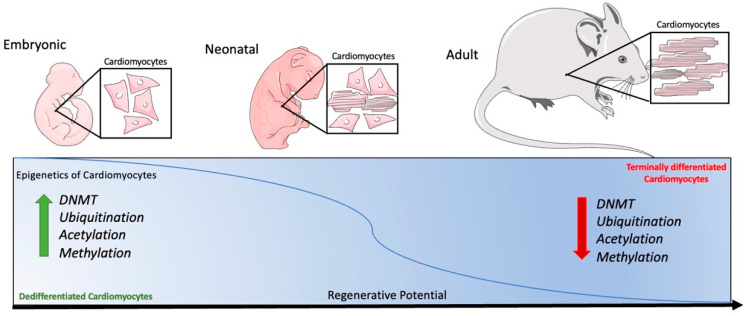
Overview of murine epigenetic regulation and cardiomyocyte regenerative potential through development.

## Data Availability

Not applicable.

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
