# Peer review of "Targeting Epigenetic Regulation of Cardiomyocytes through Development for Therapeutic Cardiac Regeneration after Heart Failure"

_ijms, 2022, doi:10.3390/ijms231911878_

Round 1
Reviewer 1 Report
It’s a very important and interesting topic, and the writing is excellent. I have some suggestions:
1, Other epigenetic modifications such as m6A RNA methylation and noncoding RNAs need to be addressed.
2, The process of cardiac differentiation and maturation may need to be described.
3, Mitochondria, which is essential to epigenetic modifications, changes dramatically during cardiac differentiation and maturation. It would be much better if the role mitochondria can be discussed.
Author Response
I would like to thank Reviewer 1 for their time, comments, and critiques. Based on the initial feedback, I have incorporated all the changes suggested, as well as updated grammatical errors and English edits.
- I have since added a section on m6A RNA methylation. This is a vital component of the review. An entire section (now 3.2) has been added with sources and explanations. The modification is also addressed in Table 1. Non-coding RNAs have also been addressed in the Conclusion, along with other RNAs.
- I have updated section 2 of the manuscript entitled "Mammalian Cardiomyocyte Development and Differentiation” to better explain and emphasize the process of cardiac cell differentiation and maturation.
- The role of epigenetics and mitochondria have also now been added in both the introductory sections as well as in the specific epigenetic modifications.
Reviewer 2 Report
In this manuscript, the researchers tried to explain about the Targeting Epigenetic Regulation of Cardiomyocytes through Development for Therapeutic Cardiac Regeneration after Heart Failure. It is interesting work and can be accepted after revision.
- The grammar errors should be checked in the whole manuscript.
- In abstract, the first few lines should be summarized.
- In introduction, the main objective has been repeated so it should be refined.
- Some recent and relevant articles may be added as hundreds of articles have been published on this topic.
- Conclusion should be refined as it is not properly written as per results
Author Response
I would like to thank Reviewer 2 for their time, comments, and critiques. Based on the initial feedback, I have incorporated all the changes suggested. Grammar errors have been fixed and the editing has been updated throughout the entire manuscript. Below addressed each of the individual comments.
- The first few lines of the abstract have been summarized.
- The main objective was repeated many times and has since been condensed and summarized, specifically in the introduction.
- More recent and relevant articles have been added to all sections. The new citations can be found in the manuscript and in the References.
- The Conclusion has also been refined and edited to better represent the manuscript.